# Exploring nonlinear dynamics in periodically driven time crystal from synchronization to chaotic motion

Alex Greilich ®[1,4] ✉, Nataliia E. Kopteva ®[1,4] ✉, Vladimir L. Korenev[2], Philipp A. Haude[1] & Manfred Bayer ®[1,3]

The coupled electron-nuclear spin system in an InGaAs semiconductor as testbed of nonlinear dynamics can develop auto-oscillations, resembling time-crystalline behavior, when continuously excited by a circularly polarized laser. We expose this system to deviations from continuous driving by periodic modulation of the excitation polarization, revealing a plethora of nonlinear phenomena that depend on modulation frequency and depth. We find ranges in which the system's oscillations are entrained with the modulation frequency. The width of these ranges depends on the polarization modulation depth, resulting in an Arnold tongue pattern. Outside the tongue, the system shows a variety of fractional subharmonic responses connected through bifurcation jets when varying the modulation frequency. Here, each branch in the frequency spectrum forms a devil's staircase. When an entrainment range is approached by going through an increasing order of bifurcations, chaotic behavior emerges. These findings can be described by an advanced model of the periodically pumped electron-nuclear spin system. We discuss the connection of the obtained results to different phases of time matter.

Nonlinear systems are characterized by significant interactions between their constituents, resulting in collective dynamics[1]. A prominent example of nonlinear effects is frequency synchronization, a phenomenon observed for living organisms, like for heartbeats or cricket chirping, as well as for nonliving systems, such as clocks and generators. Synchronization refers to the adjustment of the frequencies of autonomous systems, which periodically oscillate for constant external excitation, due to their weak interaction.

Particularly interesting is the synchronization of the system's auto-oscillations with an external periodic source, the frequency of which is close to one of the frequencies in the unperturbed harmonic spectrum. Then, even for a weak influence, significant changes occur, such as the adjustment of the circadian rhythms in organisms to the day-night cycle or the control of the pacemakers regulating heart rhythms. Despite their seeming differences, these synchronization phenomena can be understood using the common framework established for the nonlinear dynamics of complex systems[1–3].

Implementing synchronization in semiconductors is an intriguing possibility because of the application relevance of these materials, e.g., in modern electronics. Previously, we revealed highly robust, non-decaying auto-oscillations in the electron-nuclear spin system (ENSS) of a tailored semiconductor[4] as a clear signature for a continuous time crystal. The concept of time crystals was introduced in 2012 by Frank Wilczek[5,6] for a closed many-body system in thermodynamic equilibrium. While this original time crystal phase is prohibited[7–9], time crystalline behavior demonstrating spontaneous breaking of the time translation symmetry was predicted to be feasible in non-equilibrium. Such systems offer an attractive avenue for high-precision exploration of non-equilibrium states of matter.

[1]Experimentelle Physik 2, Technische Universität Dortmund, Dortmund, Germany. [2]Ioffe Institute, St. Petersburg, Russia. [3]Research Center FEMS, Technische Universität Dortmund, Dortmund, Germany. [4]These authors contributed equally: Alex Greilich, Nataliia E. Kopteva. ✉e-mail: alex.greilich@tu-dortmund.de; natalia.kopteva@tu-dortmund.de

During the last decade, two distinct scenarios have been developed: continuous (CTC) and discrete (DTC) time crystals, where the external source is acting in a steady or modulated way, respectively[10–12]. For excitation without modulation, dissipative CTCs become possible, where the continuous energy flow from an external source is transformed into oscillatory motion in so-called autonomous systems. This type of behavior has been discovered across manifold different physical systems ranging from Bose-Einstein-Condensates[13], Rubidium atom condensates[14], Erbium ion ensembles[15], strongly interacting Rydberg gases[16], light fields in photonic metamaterials[17], polariton condensates in semiconductors[18], to electrons in electrostatically defined double quantum dots[19] and mesoscopic fractional quantum Hall devices[20,21]. For modulated excitation, DTCs were experimentally confirmed in various systems by demonstrating subharmonic responses to periodic excitation.[22–31]

Here, we use the robust CTC platform implemented in a semiconductor ENSS to experimentally and theoretically explore the nonlinear dynamics achieved by deviating from continuous driving through periodic modulation of the exciting laser polarization. The deviation causes a wide variety of phenomena ranging from synchronization to chaotic motion that depend on the frequency and depth of the modulation. The synchronization becomes evident through frequency entrainment, where the ENSS synchronizes with the modulation frequency or its rational fractions. The width of the entrainment frequency bands depends on the modulation depth, leading to an Arnold tongue pattern[32,33]. Furthermore, we observe frequency bifurcation jets at the entrainment edges. Across the different ranges, each frequency branch in the system's response corresponds to a devil's staircase[32]. The entirety of the branches forms a fractal pattern showing self-similarity, confirmed by simulations. Going through an increasing number of bifurcations, we see a transition to chaotic behavior before reaching entrainment again. The entirety of different phenomena observed in a single system underscores that this system is an ideal testbed for the nonlinear dynamics of nonequilibrium systems, particularly for understanding synchronization phenomena.

It is essential to highlight the difference between the synchronization observed in nonlinear autonomous systems and another well-known phenomena, resonance and parametric resonance. The latter can be observed in periodically driven systems that do not demonstrate self-sustained oscillations. So, there is no synchronization, as the system does not have its own rhythm, and if the periodic driving is interrupted, the oscillation stops after some transition time[2].

The effect of parametric resonance or subharmonic oscillations, initially observed by Faraday in periodically driven granular material[34,35], represents a fundamental mechanism where a system oscillates at a fraction of the driving frequency. Landau and Lifshitz later formalized this phenomenon in the context of parametric resonance[36], highlighting its universal occurrence. Unlike the classical Faraday instability, our results arise in a nonlinear auto-oscillating system, demonstrating a synchronization of higher orders and offering new insights into subharmonic dynamics.

## Results
### Periodic auto-oscillations
For our studies, we have used the same ternary semiconductor structure, with which the CTC was implemented in a reservoir of In, Ga, and As nuclear spins[4]. The structural parameters are: A $10\,\mu m$ thick epilayer of $In_{0.03}Ga_{0.97}As$ is doped with Si donors providing an electron concentration of $3.9 \times 10^{16}\,cm^{-3}$ (see inset in Fig. 1a and Methods section for further details). The In atoms are incorporated to induce isotropic strain that causes a nuclear quadrupole splitting required for CTC operation. The reservoir also requires optical pumping because of its open system character. The resulting losses of angular momentum, e.g., due to nuclear dipole-dipole interaction, can be exactly compensated by circularly polarized laser excitation, which orients the localized electron spins of the donors, from which access is granted to the nuclear reservoir via the hyperfine interaction. This interaction within the electron localization volume maintains the nuclear spin polarization via flip-flop processes, establishing a robust CTC in the illuminated volume.

The photoluminescence of this structure at low temperature ($T = 6$ K) is governed by the emission from free and donor-bound excitons; see the blue curve in Fig. 1a. Using circular polarization of a pump laser at $E_{pu} = 1.579$ eV photon energy, non-zero electron spin polarization is created, which can be assessed by measuring the Faraday rotation (FR) of a linearly polarized continuous wave probe laser. The FR spectral dependence is shown by the red trace in Fig. 1a. In all further experiments, the probe energy is fixed at $E_{pr} = 1.454$ eV, corresponding to optimized experimental conditions: The FR has a local maximum with weak probe laser absorption. Measuring the Faraday rotation by the electron spins also gives us access to the nuclear polarization, which is imprinted in the electron spins through the effective nuclear magnetic field (Overhauser field).

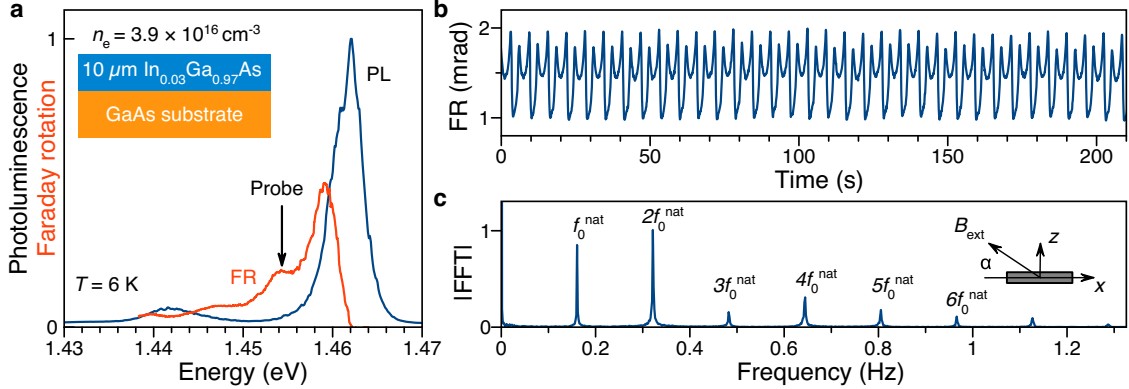

**Fig. 1 | Optical properties and auto-oscillations. a** Photoluminescence spectrum at $T = 6$ K, excited by a continuous wave diode laser at $E_{pu} = 1.579$ eV photon energy (blue line). The red trace shows the Faraday rotation spectrum, measured by tuning the photon energy of the continuous wave probe laser. The inset shows a sketch of the Si-doped $In_{0.03}Ga_{0.97}As$ layer with resident electrons concentration of $n_e = 3.9 \times 10^{16}\,cm^{-3}$ on top of a GaAs substrate. **b** Periodic auto-oscillations measured in a tilted magnetic field with components $B_x = -1$ mT and $B_z = 0.176$ mT,

corresponding to the tilt angle $\alpha = 10°$. The time trace recording started after five minutes of pumping for transients to have faded. Pump and probe photon energies: $E_{pu} = 1.579$ eV, $E_{pr} = 1.454$ eV. Pump and probe powers: $P_{pu} = 0.3$ mW, $P_{pr} = 1$ mW. **c** Fast Fourier transform of the signal in panel (**b**), recorded for 10 min. The inset shows a sketch of the experimental geometry with the magnetic field orientation relative to the sample.

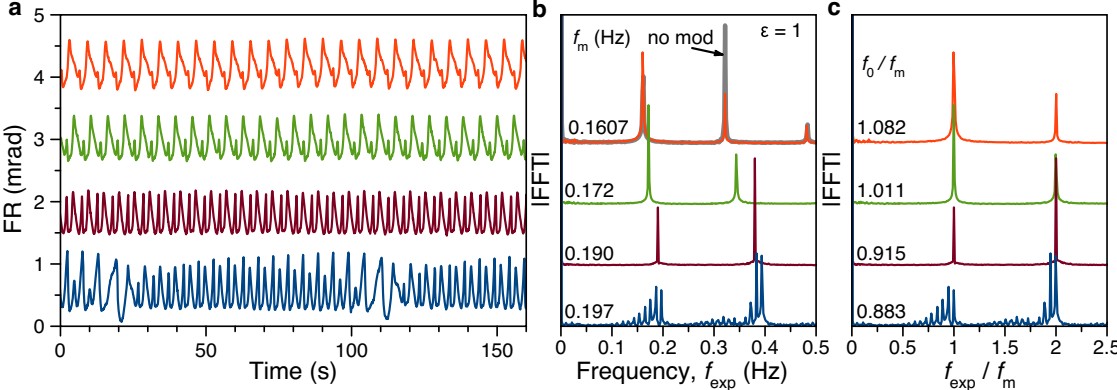

**Fig. 2 | Periodically driven ENSS. a** FR signal of periodically driven oscillations at frequencies deviating increasingly from the basic eigenfrequency, first starting with this frequency $f_m = 0.1607$ Hz (red), and then increasing to $f_m = 0.172$ Hz (green), $f_m = 0.190$ Hz (brown), $f_m = 0.197$ Hz (blue), all for $\varepsilon = 1$. The other experimental parameters are as in Fig. 1. The trace recording starts after five minutes of waiting time with the pump on to have transient effects suppressed. **b** FFT spectra of the signals from panel (**a**), recorded for 10 minutes. The gray trace reproduces the FFT of the unmodulated signal with the basic harmonic $f_0^{nat} = 0.1607$ Hz. **c** Same FFT spectra as in panel (**b**) with the frequency axis normalized by $f_m$, so that the FFT peaks are held at constant positions in the case of entrainment. The labels at the curves give $f_0/f_m$. Note that $f_0$ is affected by the average helicity value.

For specific pumping conditions (continuous, non-modulated excitation with, for example, magnetic field components $B_x = -1$ mT and $B_z = 0.176$ mT, laser powers $P_{pu} = 0.3$ mW and $P_{pr} = 1$ mW, $T = 6$ K), the ENSS demonstrates robust non-decaying auto-oscillations - a clear CTC manifestation, see Fig. 1b, as studied in detail in ref. 4. The fast Fourier-transformed (FFT) spectrum of these oscillations is shown in Fig. 1c, corresponding to a CTC structure analysis like for a space crystal, where X-ray inspection gives the Fourier transform of the electron charge density in a crystal lattice. The spectrum is harmonic with the ground (natural) frequency $f_0^{nat} = 0.1607$ Hz and the higher harmonics $n f_0^{nat}$, where $n$ is an integer.

**Periodic modulation of excitation polarization**

We apply the modulation to the pump laser polarization at the frequency $f_m$, and keep the excitation power constant. The degree of circular polarization is varied in the following way: $\rho_c = \frac{1}{2}[2 - \varepsilon(1 - \cos(2\pi f_m t))]$, with $\varepsilon$ being the modulation depth. For $\varepsilon = 0$, the light polarization is fixed at circular; for $\varepsilon = 1$, there is variation between circularly and linearly polarized pumping, while for $\varepsilon < 1$, the light is modulated between circular and elliptic polarization of a certain degree, see Supplementary Fig. 1b.

Figure 2a demonstrates exemplary time traces for $f_m$ varied starting from proximity to the $f_0^{nat}$ of the unperturbed CTC case to higher frequencies with $\varepsilon = 1$, where otherwise the same experimental conditions were applied as for the auto-oscillations in Fig. 1b. The corresponding FFT spectra in Fig. 2b, taken for time traces with 10-minute recording time, reveal a non-trivial behavior. The red trace demonstrates the case for $f_m = 0.1607$ Hz, close to the basic CTC harmonic of the autonomous ENSS. The spectral positions of the FFT peaks coincide with those of the unperturbed case shown by the gray spectrum behind the red one in Fig. 2b, but the amplitudes differ. When $f_m$ increases, the FFT harmonics shift rigidly with the applied modulation frequency. This behavior is well known from periodically driven dissipative systems, titled frequency locking[37] or entrainment[2,38]. In our case, the whole frequency spectrum $f_{exp}$ of the system is entrained with the modulation frequency $f_m$. This situation continues up to the frequency of $f_m = 0.197$ Hz, where rather abruptly a multitude of subharmonics appears; see the bottom blue spectrum in Fig. 2b. The frequency range across which the spectrum is locked to the modulation frequency shift from the unperturbed spectrum without any appearance of additional subharmonics is called the entrainment range. We normalize the FFT spectra by $f_m$ to highlight the

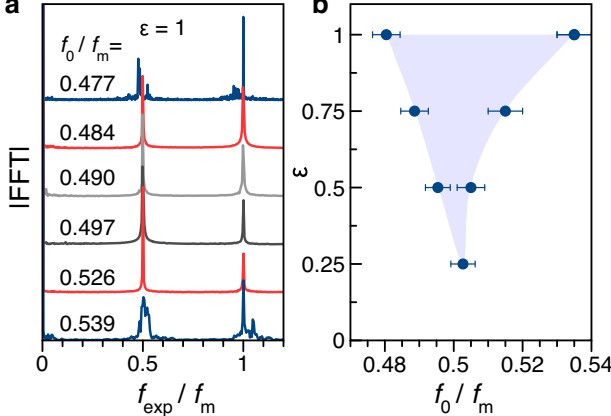

**Fig. 3 | Entrainment range and Arnold tongue. a** FFT spectra for $f_m$ varied around twice the ENSS basic harmonic ($2 f_0^{nat}$) with full modulation depth $\varepsilon = 1$. Here, the frequency scale is also normalized to the modulation frequency $f_m$. **b** Points mark the borders of the entrainment range (blue area) in dependence on the modulation depth, forming an Arnold tongue. The borders are set at the half-step size between the synchronized case with clean FFT peaks and the unsynchronized case with multiple side peaks in units of $f_0/f_m$. The error bars are correspondingly given by half-step sizes in the same units.

frequency entrainment. Then, the peaks do not change their spectral position across the entrainment range, see Fig. 2c.

It is important to note that $f_0^{nat}$ is the system's natural frequency without modulation. If modulation is applied, the pump's average helicity is changed, renormalizing the natural frequency to $f_0$.

In the Supplementary Fig. 2, we additionally show time traces and corresponding FFT spectra for $f_m$ that match several rational fractions of $f_0$ to $f_m$ in the range from $f_0/f_m = 1/2$ to $f_0/f_m = 1$. The FFT spectra demonstrate subharmonic frequencies, which divide the spectra into several equidistant intervals. The number of these intervals for $f_{exp}/f_m \leq 1$ is equal to the denominator of $f_0/f_m$. Close to these rational fractions, one observes frequency entrainment. As an example, Fig. 3a presents the FFT spectra for $f_0/f_m$ varied around 1/2, from 0.477 to 0.539, for $\varepsilon = 1$, evidencing a range of entrainment with two FFT peaks in the shown frequency interval: at the modulation frequency $f_m \approx 2 f_0^{nat}$, close to the second harmonic of the unmodulated ENSS, and at half of the modulation frequency representing a subharmonic

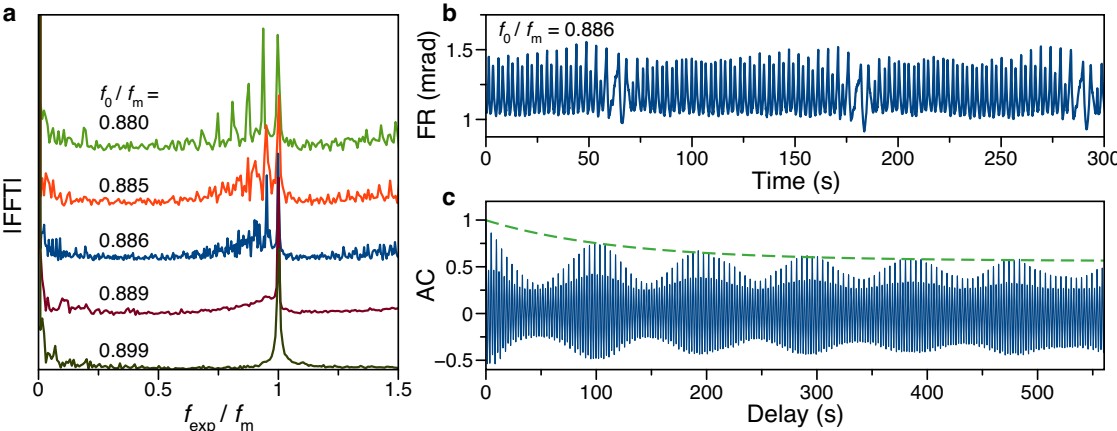

**Fig. 4 | Chaos at edge of entrainment range. a** FFT spectra for $f_m$ varied close to the ENSS basic harmonic $f_0^{nat}$ for full modulation depth $\varepsilon = 1$. Again, the abscissa is normalized to the modulation frequency. **b** Faraday rotation signal of periodically driven oscillations with frequency $f_0/f_m = 0.886$. **c** Autocorrelation (AC) function calculated for the signal in (**b**). The green dashed line highlights the AC decay.

response. As discussed in the introductory paragraphs, observing a stable subharmonic response with entrainment to the modulation frequency is generally considered a signature of a DTC state, as will be discussed further below. The blue FFTs with multiple subharmonics in Fig. 3a indicate the boundaries of the entrainment range.

At the edges of all entrainment ranges, the FFT spectra consist of multiple peaks around the main harmonics. Exemplarily, we discuss the edge at $f_0/f_m = 0.883$, shown in Fig. 2c, and analyze the spectra taken in small frequency steps to characterize the transition across the edge carefully, see Fig. 4a. We chose this edge due to a broader range on the low-frequency side of modulation that is not obscured by additional subharmonics in the entrained state which reduces the amplitudes of contributing frequencies.

Here, at the very edge of the transition to entrainment, the number of FFT peaks drastically increases, see the spectrum for $f_0/f_m = 0.886$ with the corresponding time trace in Fig. 4b. The merging FFT peaks are a signature of chaotic oscillations. To confirm this, we perform several chaos tests established in the literature.

To that end, a 30-min recording time trace is measured and analyzed. The auto-correlation function shows slowly decaying beats with increasing delay. The beats appear due to the superposition of oscillating contributions with frequencies $f_0$, $f_m$, and commensurate harmonics of both of these[39]. The slow decay is an indication of deviation from periodic behavior. A nonlinear time series analysis gives the positive maximal Lyapunov exponent $\lambda_{max} = 0.06$[40], evidencing an exponential deviation of closely spaced initial states with time, and a non-integer correlation dimension of $D_2 = 2.5$[41], confirming the chaotic behavior[42] (see Supplementary Fig. 3a–c). For comparison, we also give as an example the same quantities for the synchronization plateau at $f_0/f_m = 2/3$, resulting in $\lambda_{max} = 0$ and $D_2 = 1$, see the Supplementary Fig. 3d–f, confirming a near ideally periodic behavior. Additionally, there are ranges between the synchronization plateaus where the correlation dimension has the value of $D_2 = 2$, which represents quasi-periodicity with an irrational ratio among the observed frequencies (see the Supplementary Fig. 3g–i) and corresponds to an intermediate situation where the system is neither periodic nor chaotic[42].

Finally, to reveal the complete picture of the ENSS response to frequency modulation, we provide FFT spectra for a range of $f_m$ varied from the basic harmonic $f_0^{nat}$ up to slightly more than twice this frequency. Here, we use the inverted frequency scale again so that the abscissa interval covers $0.45 \leq f_0/f_m \leq 1$, for $\varepsilon = 1$. The resulting measurement series is shown as a color map in Fig. 5a (and in Supplementary Fig. 7 in stretched format for more details) and demonstrates

a multitude of frequencies in the subharmonic regime below the applied modulation frequency $f_m$.

We want to highlight some points introduced before to interpret the observed map. The horizontal plateaus of finite width seen at rational values of $f_0/f_m$ (see top axis in Fig. 5a) represent entrainment ranges. The length of the plateaus decreases with increasing denominator. So, the plateaus at the ratios 1/2, 3/5, 2/3, 3/4, and 4/5 have lengths of 0.059, 0.012, 0.018, 0.013, and 0.010 ± 0.001 in units of $f_0/f_m$, correspondingly. Leaving an entrainment range by moving to smaller $f_m$, we observe a splitting of each frequency into many branches, comprising bifurcation jets[38]. Moving further and approaching the next entrainment range, the number of bifurcations continuously increases and becomes so high that eventually, chaotic behavior emerges through the coupling between the different resonances caused by the nonlinearities[32].

## Model of periodically modulated ENSS

To obtain additional insight into the experimentally discovered behavior, we have generalized the model described in refs. 4,43 for an autonomous system by adding the periodic force variation provided by polarization modulation at frequencies close to the intrinsic auto-oscillation. We briefly repeat the basic features: The circularly polarized pump excitation orients the donor electron spins, which subsequently polarize the nuclear spin system via the hyperfine interaction[44]. The Overhauser field of the polarized nuclear spins $\mathbf{B}_N$, in general, is oriented not parallel to the average electron spin $\mathbf{S}$, so that an electron spin precesses about $\mathbf{B}_N$, causing a variation of $\mathbf{S}$. Thus, in the strongly coupled nonlinear ENSS system, the electron spins and the Overhauser field are mutually interdependent via their magnitude and direction. Then, for continuous pumping, the dynamic regime of self-sustained auto-oscillations appears under specific conditions.

Due to the short electron spin lifetime ($T_s \sim 1\,\mu s$) compared to the longitudinal nuclear spin relaxation time ($T_N \sim 1\,s$), the electron spin ($\mathbf{S}$) is described by the solution of the stationary Bloch equation accounting for the sum of the external magnetic field ($\mathbf{B}_{ext}$) and the Overhauser field ($\mathbf{B}_N$):

$$\mathbf{S} = \mathbf{S}_0 + \frac{\mu_B g T_s}{\hbar}(\mathbf{B}_{ext} + \mathbf{B}_N) \times \mathbf{S}. \tag{1}$$

Here, $\mathbf{S}_0$ is the average electron spin polarization induced by the pump without magnetic field, $\mu_B$ is the Bohr magneton, $\hbar$ is the reduced Planck constant, and $g$ is the electron $g$-factor.

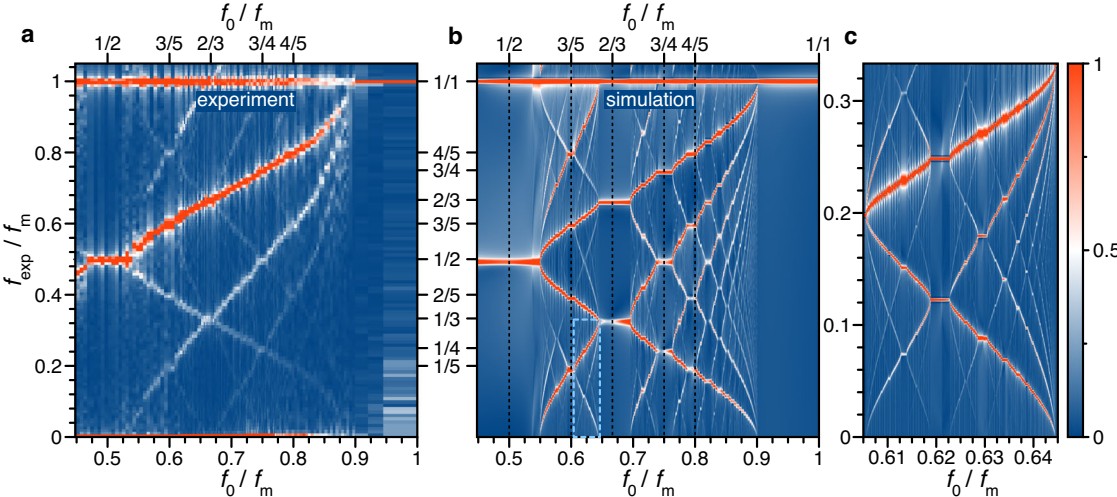

**Fig. 5 | Entrainment, bifurcations, and fractal structure. a** Contour plot of experimental FFT spectra with the frequency axis $f_{exp}$ normalized by the modulation frequency $f_m$ as a function of inverse modulation frequency, multiplied by the basic harmonic frequency. $\varepsilon = 1$. For each FFT, time traces of 10 minutes of recording time are used. **b** Same as (**a**), but with FFT spectra from simulations, using the parameters from Fig. 2c and $\varepsilon_{sim} = 0.5$. To highlight the fractal nature of the signal, panel (**c**) shows a zoom of the area marked by the light blue box shown in (**b**). The scale on the right shows the color scheme for the normalized amplitude of the contour maps.

The precession of the electron spin polarization about the total magnetic field ($\mathbf{B}_{ext} + \mathbf{B}_N$) changes the Overhauser field in time according to[44,45]:

$$\frac{d\mathbf{B}_N}{dt} = -\frac{1}{T_N}(\mathbf{B}_N - \hat{a}\mathbf{S}), \tag{2}$$

where $\hat{a}$ is a second-rank tensor describing the process of dynamic nuclear polarization. The model suggests that $\hat{a}\mathbf{S}$ is a linear function of $\mathbf{S}$. The tensor has been simplified in the lowest order approximation (for details, see Methods section, ref. 4 and corresponding Supplementary Material). This allowed us to simulate the CTC auto-oscillations, as shown in ref. 4. For the calculations in this paper, we use the same experimental parameters as in the previous work: $\alpha = 10°$, $B_x = -1$ mT, effective magnetic fields of $a_N = 20$ mT and $b_N = 21$ mT, and nuclear spin relaxation time of $T_N = 0.5$ s.

We extend the model by implementing the modulation of laser polarization, leading to the periodicity of $\mathbf{S}_0$ with the same frequency: $\mathbf{S}_0$ follows the degree of circular polarization $\rho_c$ so that we substitute it in Eq. (1) by:

$$\mathbf{S}_{0,m} = \frac{\mathbf{S}_0}{2}[2 - \varepsilon_{sim}(1 - \cos(2\pi f_m t))], \tag{3}$$

where $\varepsilon_{sim}$ is the modulation depth in the simulation, allowing a deviation from the experimental $\varepsilon$. The analysis of the dynamical Eqs. (1)–(3) reveals several new results, which are described below.

We simulate the electron spin in the presence of the nuclear field to obtain the measured signal for varying modulation frequency $f_m$ and calculate the dependence of the FFT using 20-minute time traces, which evolve to form both frequency entrainment ranges and complex bifurcation patterns between them, as presented in Fig. 5b. The Supplementary Fig. 4 additionally presents the simulated maps for different depths of modulation $\varepsilon_{sim}$. Overall, we find good agreement between calculation and experiment, but with an adjusted value of $\varepsilon_{sim} = 0.5$ compared to $\varepsilon = 1$ in the experiment, which can be explained by several factors, including strongly non-resonant pump excitation, resulting in partial electron spin relaxation.

Again, we plot the spectra in a contour as a function of the modulation rate $f_0/f_m$[33]. As mentioned, using this coordinate, the

entrainment branches are identified as horizontal lines. An additional advantage of this presentation is the possibility to introduce a characteristic parameter, the so-called effective winding number $w = f_{circ}/f_m = T_m/T_{circ}$, which is the ratio of the modulation period $T_m$ to the time $T_{circ}$ that the ENSS takes to return to its initial point during motion on a closed periodic trajectory (limit cycle). This quantity was introduced as a simplified version of the winding number in the circle map representation of refs. 33,42.

To understand its connection with the observed spectra, let us consider the entrainment plateau in Fig. 5b around $f_0/f_m = 1$ on the x-axis and at $f_{exp}/f_m = 1$ on the y-axis. In this case, the system undergoes one full revolution along the limit cycle during one period $T_m$ of modulation, giving the effective winding number $w = 1$. Further, for the entrainment plateau around $f_0/f_m = 1/2$ on the x-axis and at $f_{exp}/f_m = 1/2$ on the y-axis the system undergoes one full revolution along the limit cycle during two periods $T_m$ of modulation, giving the effective winding number $w = 1/2$. Larger values of $f_0/f_m > 1/2$ associated with entrainment plateaus have multiple frequency peaks for $f_{exp}/f_m \leq 1$, where only the lowest frequency corresponds to $w = f_{circ}/f_m$.

Our calculation shows that near resonances $f_0/f_m = M/N$ the effective winding number $w = 1/N$, where $M$, $N$ are mutually prime (coprime) natural numbers[33]. The Supplementary Fig. 2a demonstrates the period $T_{circ}$ for several cases, representing the lowest harmonic frequency in the corresponding FFT spectra in the Supplementary Fig. 2b. We additionally show several examples of simulated spin trajectories giving all vector components, which highlight the spin evolution for the cases $T_{circ} = T_m$, $2T_m$, $4T_m$, see the Supplementary Fig. 5a–c, respectively. In the measured spectra of Fig. 5a, there are also other harmonics, so that the whole series of observed frequencies is given by $f_{exp}/f_m = K/N$, for $K = 1, 2, \ldots$.

Remarkably, the entrainment ranges associated with the numbers $f_{exp}/f_m = M/N = 1/N$ or $M/N = (N-1)/N$ in dependence of $f_0/f_m$ form devil's staircases[46,47]. The step width in each staircase becomes the smaller, the larger the denominator $N$[32], following the so-called Farey tree sequence, eventually leading to a chaotic state[42]. Our simulation shows that such ranges with a very high number of bifurcations appear close to each transition to synchronization, see Fig. 5b, c. Due to the redistribution of the frequency components between multiple subharmonics, these are much harder to observe experimentally.

The observed devil's staircases evidence the self-similar character of fractal structures, i.e., each detailed section, no matter how small, contains the same features as the whole picture. Figure 5c shows a zoom of the simulated part marked by the light blue-dashed box in Fig. 5b, basically reproducing the large area and confirming the self-similarity. A similar behavior, in terms of devil's staircase structures, was recently demonstrated for frequency-locked breathers in ultrafast lasers[48] and theoretically proposed for the optomechanical locking in driven coupled polariton condensates[49], confirming the universal nature of the observed non-linear phenomena.

Furthermore, reducing the modulation depth $\varepsilon$ leads to the narrowing of the observed entrainment ranges. To confirm this, we determine the edges of the experimentally observed entrainment range close to $f_0/f_m = 1/2$ and plot it as a function of $\varepsilon$ by the symbols in the phase diagram in Fig. 3b. Below $\varepsilon = 0.25$, the entrainment vanishes so that the FFT spectra resemble that of the unmodulated ENSS, as the deviations from the circular pumping become too small. This demonstrates the remarkable stability of the ENSS for CTC operation with respect to variations in the laser helicity ($\varepsilon < 0.2$). The dependence of the width of the entrainment range on the modulation depth results in the blue shaded area in Fig. 3b, which is known as Arnold tongue[2,50]. Additionally, the Supplementary Fig. 6 shows the dependence of the auto-oscillations on the temporally constant degree of pump polarization, having maximal amplitude for circular polarization while dropping to zero for linear polarization.

In ref. 32, the authors studied the scaling law underlying the Arnold tongues. Similar to their results, the observed fractal structure, in our case, is related to the structure of the self-similar Cantor set. Using the experimentally observed Farey tree sequence with major plateaus and gaps between them, we arrive at the fractal dimension value of $D' = 0.852 \pm 0.021$, while the simulated data, with the possibility to zoom into the staircase structure, delivers $D' = 0.853 \pm 0.002$, see Methods. These values are close to 0.87 expected for the complete devil's staircase[46] and align with the universal properties of mode-locking transitions and the devil's staircase structure seen in dissipative systems[51] as well as in refs. 48,52,53. This suggests that the ENSS likely belongs to a broader universality class, sharing convergence and scaling properties akin to those found in circle maps and forced oscillators.

To round out the picture, the Supplementary Fig. 4 shows simulated color maps for decreasing $\varepsilon_{sim}$, demonstrating how in parallel the entrainment ranges narrow and the bifurcation structures diminish, leading to the disappearance of the devil's staircases. They also confirm the Arnold tongue structure in Fig. 3b. On the other hand, with increasing modulation depth ($\varepsilon_{sim} > 0.5$), the entrainment ranges (or neighboring Arnold tongues) start to overlap, also leaving no space for bifurcations. This situation cannot be reached for pump polarization modulation in the case of non-resonant excitation, but can be realized for modulation of another pump parameter, namely the pump power, as will be analyzed elsewhere.

## Discussion

Experimental results are well described within the framework of nonlinear physics, which can and needs to be connected further to the physics of time crystals in nonlinear systems. As discussed in the introductory paragraphs, observing a stable subharmonic response with entrainment to the modulation across a finite frequency range, like for the ENSS here, is generally related to a DTC state. The entrainment range represents the area of stability for such a state. Further, the emergence of multiple subharmonic resonances at specific fractions $f_0/f_m$, as observed in the Supplementary Fig. 2b, has been discussed in terms of fractional and higher-order DTC states[54–59]. In this regard, the CTC state, in our case, undergoes the transition to a DTC phase, where the subharmonic response occurs on rational fractions of the modulation frequency $f_m$, as well described by our model.

Recently, a similar transition was demonstrated for an atom-cavity system[60]. Considering all these findings, our DTC phase belongs to a specific class of nonlinear dynamic systems.

The discussion in terms of time crystals is generally applicable to strictly periodic phenomena in autonomous systems on a limit cycle and non-autonomous systems in the synchronization regime. Our system also features aperiodic modes of nonlinear systems, such as chaotic oscillations and fractal structures with quasi-periodic oscillations in the subharmonic range. The former system phase may be called melted time crystal or time glass and the latter phase time quasi-crystal, similar to recent experimental work on a strongly interacting spin ensemble in diamond[61]. However, so far, the definition of time matter is not fully clear-cut within the general frame of dynamics of nonlinear autonomous and non-autonomous oscillating systems or whether it even goes beyond that frame.

The connection between our findings and those in Rydberg gases[57–59], Bose-Einstein-Condensates of Rubidium atoms[60], exciton-polariton systems[49], and spins in diamond[61], underscores the universal mechanisms underlying time-crystalline behavior across different physical systems. Our solid-state platform offers new opportunities for further studies of nonlinear dynamics.

The phenomena studied in this work are primarily driven by the local dynamics arising from nonlinear feedback between the electron and nuclear spins within an ensemble of donors. However, the spatial coupling between donors would introduce a degree of nonlocality, which can lead to spatial synchronization across neighboring donors. While our current analysis focuses on temporal dynamics, future work will explore potential spatial patterns in terms of partial synchronization[62–64] or instabilities in the chaotic regime, similar to those described in spatially extended systems[65,66].

Furthermore, the use of a semiclassical model in our case is justified by the large ensemble of nuclear spins (more than $10^5$ per donor) and multiple orders of magnitude of time-scale separation between the electron and nuclear spin dynamics. However, one could consider detecting quantum correlations among separate single donors, which would require high spatial resolution and a diluted nature of Si doping for their experimental investigation.

Finally, the initially observed, unmodulated CTC state of the ENSS represents a limit cycle in the phase space. The feedback strength in this nonlinear system, which keeps it on a limit cycle, determines the oscillation stability (i.e., the quality factor). The results presented here can also be seen from a different perspective, where the external quality factor is mapped on the ENSS through synchronization. One may envision a situation where a highly stable external frequency generator stabilizes the ENSS oscillations, as is the case of quartz resonators in atomic clocks. Such an ultra-stable macroscopic state may inspire new applications in nonlinear physics and quantum technology.

## Methods
### Setup

Reference 4 and the corresponding supplementary information describe our sample and the experimental setup in all details. Here, we reiterate the most relevant parts. We use a continuous wave laser diode emitting at 1.579 eV photon energy (785 nm wavelength) as a pump laser, which is then routed through an electro-optical modulator. It changes the phase between the two orthogonally linear polarized light components of the pump in a sinusoidal way, resulting in the desired change of light polarization. The linearly polarized probe laser is created by a continuous wave Ti:Sapphire ring-laser and is fixed at 1.454 eV photon energy (852.63 nm wavelength). Both lasers are combined on a non-polarizing beam splitter and focused by a single lens onto the sample. The pump laser is completely absorbed by the GaAs substrate of the sample, while the probe laser is transmitted through it and then analyzed by a polarization bridge, which consists of a half-wave plate

and a Wollaston prism. A balanced photodiode is used to measure the Faraday rotation of the probe's linear polarized plane, see Supplementary Fig. 1a. The sample is mounted in a helium flow cryostat at temperature $T = 6\,K$ in the center of two orthogonal pairs of electromagnet coils, generating the magnetic field components $B_x$ and $B_z$.

## Details on data processing

To detect the position of $f_0$ in the modulation case, we set the modulation frequency $f_m$ to about $3.5 f_0^{nat}$. In this way, the position of $f_0$ is unaffected by the proximity of the modulating frequency.

## Details on simulation model

The quadrupole unperturbed nuclear spin sublevels create the Overhauser field $\mathbf{B}_N^0$ as in pure, unstressed GaAs. $\mathbf{B}_N^0$ is aligned along the external magnetic field and compensates for the Zeeman splitting of the electrons in the external magnetic field. Due to the strong deformation caused by the indium incorporation, the spin of the $i$-th nucleus is oriented along the main local axis $\mathbf{n}_i$ of the tensor describing the quadrupole interaction rather than along the external magnetic field. The contribution of these nuclei to the total Overhauser field is $\mathbf{B}_Q = \sum_i a_i(\mathbf{Sn}_i)\mathbf{n}_i$, where the summation is carried out over all quadrupole perturbed nuclei within the electron localization volume around a donor. For an isotropic distribution of the axes, the field can be written as $\mathbf{B}_Q = a_N\mathbf{S}$. Therefore, $\hat{a}$ can be reduced to the simplified form: $\hat{a}\mathbf{S} = \mathbf{B}_Q + \mathbf{B}_N^0 = a_N\mathbf{S} + b_N(\mathbf{Sh})\mathbf{h}$, where $b_N$ is the parameter of the hyperfine interaction between the electrons and the nuclei, and $\mathbf{h}$ is the unit vector of the externally applied magnetic field. The tensor components are: $\hat{a}_{\alpha\beta} = a_N\delta_{\alpha\beta} + b_N h_\alpha h_\beta$, with $\alpha, \beta = x, y, z$ coordinates.

## Fractal dimension

We select out two arbitrary neighboring rotation numbers $p'/q'$ and $p''/q''$ and measure the length of the gap $S$ in between. We know from the Farey tree structure that the largest synchronization interval in between belongs to the rotation number $(p' + p'')/(q' + q'')$. The gaps between the new interval and the two previous ones are denoted $S'$ and $S''$. According to ref. [67], an approximation $D'$ for the fractal dimension $D$ can be determined from the relation: $(S'/S)^{D'} + (S''/S)^{D'} = 1$. A better approximation is given for the dimensions by zooming into the steps structure, which is possible using the simulations. In this case, using three successive plateaus, we determine the $S_n$, $S_n'$, $S_n''$, and use: $D' = \lim_{n\to\infty} D_n$ with $(S_n'/S_n)^{D_n} + (S_n''/S_n)^{D_n} = 1$[68].

## Data availability

The data on which the plots in this paper are based and other findings of this study are available from the corresponding authors upon request.

## Code availability

The code on which the calculations within this paper are based, as well as other findings of this study, are available from the corresponding authors upon request.

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

## Acknowledgements

The authors are thankful to D. R. Yakovlev for fruitful discussions. A.G. and M.B. acknowledge support by the BMBF project QR.X (Contract No.16KISQ011). The Resource Center "Nanophotonics" of Saint-Petersburg State University provided the epilayer sample.

## Author contributions

A.G. and N.E.K. contributed equally to this paper. A.G. built the experimental apparatus and performed the measurements. P.A.H. contributed to the fine-step measurements of the contour map in Fig. 5a. N.E.K. and A.G. analyzed the data. N.E.K. and V.L.K. provided the theoretical description. All authors contributed to the interpretation of the data. N.E.K., V.L.K., and A.G. wrote the manuscript in close consultation with M.B.

## Funding

## Competing interests

The authors declare no competing interests.
