## [Transparent Peer Review file · Nature Communications]

Exploring nonlinear dynamics in periodically driven time crystal from synchronization to chaotic motion

Corresponding Author: Dr Alex Greilich

Version 0:

Reviewer comments:

Reviewer #1

(Remarks to the Author)

Please see the attached document for detailed comments.

Reviewer #2

(Remarks to the Author)

In the manuscript entitled "Exploring nonlinear dynamics in periodically driven time crystal: from synchronized to chaotic motion," the authors study deviations of the coupled electron-nuclear spin system in an InGaAs semiconductor from continuous driving by periodic modulation of the excitation polarization. The authors reveal several phenomena that depend on modulation frequency and depth. In particular, the width of these ranges depends on the polarization modulation depth, resulting in Arnold Tongues.

This work is interesting, but in my opinion, it is a natural continuation of previous works by the authors, so it seems to me that it is not within the scope of the journal in the sense that it aims to represent important advances of significance for specialists within each field. I then ask the authors to justify better why they believe the work merits publication in a high-impact journal. Furthermore, I call on you to consider the following points.

*.- In the introduction, the authors spend large passages of the text discussing the phenomenon of oscillator synchronization; however, when the system is forced periodically, other mechanisms may be relevant, such as resonance and parametric resonance. In particular, the latter presents Arnold's tongues. I recommend that this point be clarified.

*.- Subhormonal instabilities correspond to the emergence of half-frequencies [initially studied by M. Faraday, Philos. Trans. R. Soc. London 52, 319 (1831), and see Landau L. D. and Lifshitz E. M., Mechanics, (Course of Theoretical Physics, Pergamon Press, 1976)], this should be better detailed in the text.

*.- On the other hand, Figures 2b, 2c, and 4a show that complexity emerges from the quasi-periodic path. However, the analysis presented is not conclusive.

*.- It needs to be clarified whether the phenomenon studied is local or is a spatially extended phenomenon where spatial behaviors can be observed. For example, chaotic behaviors are known through quasi-periodic routing [M.G. Clerc, and N. Verschuere, Phys. Rev. E. 88, 052916 (2013)]. Comment on this point.

Minor comment

*.- The references [3] and [29] are the same.

Reviewer #3

(Remarks to the Author)

The article by A. Greilich et al presents a study of a classical, non-linear system established by an electron spin interacting with nuclear spins - all in a semiconductor structure accessed optically. The authors build up on previous work, which revealed the presence of long-lived auto-oscillations in electronic polarization (considered a continuous time crystal). The new results presented in this article stem from periodically modulating the polarization of a laser that orients the electron spin along a fixed axis. A number of fascinating phenomena arise due to feedback between the electron and nuclear spins; for certain parameters the system exhibits synchronization ('entrainment'), and it transitions into chaos as the parameters are varied. The growth of the entrainment range with an increase in modulation depth is a particularly exciting finding. The data are backed by an appropriate theoretical analysis, and the article is very well-written and cohesive.

I definitely think that the article fits well into Nature Communications and that it will attract a lot of attention from physicists from various fields. I am supportive of a publication, but I think some revisions are in place. This is because certain claims made in the article can lead to a backlash, as well as because the microscopic model is not completely clear. I would be thankful if the authors could address the following points:

1) Equation 2 describes the evolution of the Overhauser field in terms of the electronic spin. It introduces a tensor of unspecified form, and a claim is made that $\hat{a}\mathbf{S}$ is a linear function of \mathbf{S} . Should this be understood as $\hat{a}\mathbf{S}=a\mathbf{S}$ where a is a scalar? Or is \hat{a} still a tensor, but a one which has components independent of \mathbf{S} ? I believe details are important here. \hat{a} certainly should not be a scalar: without strain, in a frame rotating with the electronic precession frequency, the hyperfine tensor averages out to a matrix with a single non-zero component, reducing the hyperfine interaction between the electron and nuclei from 'Heisenberg' to 'Ising' type. As authors know, the strain is the key ingredient here, as it introduces the effective non-collinear components to the hyperfine tensor, which couple z-components of the electronic polarization with x,y-components of the nuclear polarization. These interactions allow for the polarization transfer between the electron and the nuclei, and are therefore important to elaborate on. Moreover, as written in the manuscript 'The Overhauser field of the polarized nuclear spins, in general, is oriented not parallel to the average electron spin' which would not be consistent with Equation 2 if \hat{a} was a scalar quantity. It would be great if authors presented the complete form of \hat{a} at least in methods. This also goes for all other parameters used in the model. It's impossible to reproduce the numerics without these.

2) Certain features of Fig. 5b are not reproduced in Fig. 5a. As an experimentalist, I appreciate how demanding task it was to collect the data behind Fig. 5a, and that improving the resolution along y-axis would force one to collect the data for extremely long times. However, that doesn't apply for a window of $f_0/f_m \in [0.65, 0.7]$: and there is a clear deviation of the data and simulation in that window. I believe without it, it is a little bit of a stretch to claim the observation of 'devil's staircase' structure (lines 65-68); surely it is supported by simulation, but not exactly by the data. Moreover, Fig. 5 misses a colorbar.

3) In the final paragraph of the manuscript (lines 432-434), a statement is made that 'This state can be seen as a stable internal clock that can be implemented in a semiconductor chip'. The ENSS itself is not a good frequency standard; its model parameters will vary from sample to sample, and the frequency stability of the auto-oscillations is not really explored here (nor there is any reason to believe it would be comparable to technologically relevant devices). It is therefore misleading to suggest that such semiconductor chip could realize a 'stable internal clock'. The interesting bit is that ENSS can adopt a frequency of an external frequency generator – which I fully agree with. However, I find the idea to bring it up as a metrologically-relevant outlook a little bit odd. Firstly, the modulation frequencies to which the system locks are extremely low; resolving any frequency difference in a metrological scenario would take a prohibitively long time. Could authors be a little bit more specific which metrological applications do they have in mind? It is also okay to take down the relevance of this system to metrology - it will not diminish the importance of this article..

4) The system under study is fascinating, and it looks like there could be a number of in-depth experimental studies of Chaos and non-linear dynamics performed with it in the near future. I think the article would gain from some outlooks which hint towards what's going to be explored next, some comments that draw a big picture. Beyond being a playground for chaos, it would be great to see this platform become technologically relevant. The authors suggested metrology, which I do not think is a viable route forward, but maybe there are other potential applications?

5) As a huge fan of renormalization, I have to ask this out of pure curiosity (by no means do I urge authors to include this discussion in the article): cascades of bifurcations usually exhibit convergence properties described by Feigenbaum constants (at least in unimodal maps) - what's expected in that respect from the ENSS under study?

Version 1:

Reviewer comments:

Reviewer #1

(Remarks to the Author)

I thank the authors for the detailed responses to my comments and for careful revisions of the manuscript. With this, I can

recommend publication of the paper in Nature Communications.

P.S., maybe one more comment on Q5: The notation \bar{f}_0 is suggested for the frequency under the averaged pumping. Since the authors want to keep the notation f_0 for such an averaged value, they can also choose another more suitable notation for the natural frequency, e.g., f_{0n} ("n" for "natural") or f_{0i} ("i" for "intrinsic").

Reviewer #2

(Remarks to the Author)

In the revised version, the authors have incorporated all comments from the previous peer review reports and adequately addressed most comments and suggestions. In particular, the authors more adequately justify using a semi-classical approach for their study. The disappearance of the limit cycle is clarified in the supplementary material. The state of the art has been significantly improved. Some technical aspects of the experiment, such as the power used for the light beam, have been clarified. The difference between synchronization and sustained oscillations has been better clarified. The parametric instabilities have also been clarified.

My biggest difficulty is the fulfillment of the Nature Communications criteria. The authors present a work supported by previous work, where the main novelty is considering a new platform based on a solid state of matter and its possible applications. The possibility of new applications undoubtedly gives interest to the present work on a phenomenon well-known to the nonlinear science community. Overall, the quality of the paper has improved significantly, and it is therefore recommended for publication.

Reviewer #3

(Remarks to the Author)

The authors have responded satisfyingly to all the points I raised, and the quality of the manuscript has improved. I recommend its publication in its current form.

Response to reviewers of NCOMMS-24-45342-T, "Exploring nonlinear dynamics in periodically driven time crystal: from synchronized to chaotic motion", by A. Greilich *et al.*

First and foremost, we would like to thank the Reviewers for evaluating the manuscript. Their comments and suggestions have allowed us to clarify essential questions. Please find below a point-by-point response to all comments. The changes in the main text as well as in the supplementary materials, are marked in blue color.

Reviewer #1 (Remarks to the Author):

The manuscript "Exploring nonlinear dynamics in periodically driven time crystal: from synchronized to chaotic motion" studies optical response of a periodically driven, nonlinear solid-state system. The study is based on their previous work [Nat. Phys. 20, 631 (2024)], which identifies a persistent auto-oscillation of the polarization of the probe field passing through an InGaAs semiconductor subjected to a continuous external driving. The generalization to the periodic driving regime gives rise to a number of new phenomena as key novelties of the current work, e.g., the frequency entrainment to fractional values of the driving frequency, which can be regarded as a discrete time crystal (DTC).

The authors carry out a systematic investigation of these new phenomena. They measure the time trace of the signal by varying the frequency and the depth of the periodic modulation, and analyze the results in terms of the Fourier spectrum, Arnold tongue, and bifurcation diagram. These results are of interest to the physics community, as well as to broader readers in complex systems, nonlinear sciences, and electronics. Therefore, I tend to think that the work is suitable for publication in Nat. Commun. However, before I can make the final decision, there are several critical points to be clarified and further elaborated on.

The detailed comments and questions are listed below:

R#1 1). The observed nonlinear dynamics is essentially classical, so I am wondering if the system can be extended to the quantum regime and how, e.g., what is the role of quantum correlation and entanglement and how to measure them? The authors develop a phenomenological model to describe the nonlinear spin dynamics, which has a nice agreement with the experiment. Is it also possible to develop a first-principle, fully quantized many-body model, even it may not be easy to solve? In short, I think it would be better to describe how to go beyond the classical regime, as conventional bifurcations and chaos are well understood in nonlinear sciences.

Reply: We thank the Reviewer for these crucial questions. We would like to start with a justification of the used semiclassical approach. It should be valid due to:

- Large number of involved nuclear spins ($\sim 10^5$ per donor site):
 - a. The ensemble of nuclear spins creates an effective magnetic field that can be treated as classical variable.
 - b. Quantum fluctuations are largely averaged out, allowing a deterministic treatment of the effective fields.
- Time-scale separation:
 - a. Nuclear spins evolve much more slowly (seconds) than electron spins (microseconds)
 - b. This separation could allow the electron spin dynamics to be treated quantum mechanically, while the nuclear spin dynamics can be treated semiclassically.
- The observables, such as the Faraday rotation of light by the electron spin polarization, depend on the averaged effects of the nuclear spins rather than their individual quantum states.
- The semiclassical model captures the essential electron-nuclei feedback mechanism and the dynamics of the Overhauser field. Thus, it bridges the quantum mechanical effects and their macroscopic manifestations without the need of full quantum calculations.

While theoretically possible, the fully quantum mechanical description of 10^5 nuclear spins interacting with a single donor-bound electron spin via hyperfine interaction would be highly complex and computationally intractable. Furthermore, the quadrupole splitting would complicate the formulation of the spin Hamiltonian and its solutions. Finally, the nonlinearity in the feedback would make the dynamics highly sensitive to initial conditions and external parameters.

As to the question about quantum correlations and the ways to measure them, we think an experimentalist would face the following challenges: measuring quantum correlations in a system with so many nuclear spins is quite complex; it would be quite challenging to make precise measurements of such a complex state due to decoherence; additionally, the spatial extension of the system leads to nonlocality due to spin diffusion.

On the other hand, one can speculate by considering the following potential experiments:

- Oscillations in the Faraday rotation or other observables may indirectly indicate entanglement if their dynamics cannot be explained classically. However, this requires a very high precision of the present semiclassical model, which cannot be reached currently.
- Detect quantum correlations among the donors by probing the spatial extent of synchronized oscillations. This would require isolating two separate donor centers using advanced optical or magnetic resonance techniques. Further, it would require single donor isolation, which is also not possible at the current stage.

We have included a following sentence in our paper: *“The use of a semiclassical model in our case is justified by the large ensemble of nuclear spins (more than 10^5 per donor) and multiple orders of magnitude of time-scale separation between electron and nuclear spin dynamics. However, one could consider detecting quantum correlations among separate single donors, which would require high spatial resolution and a diluted Si doping for their experimental investigation.”*

R#1 2). Related to comment 1, I noticed that the sample is already put in a low-temperature environment, where the quantum mechanical effect should play a role. Does the system have a critical temperature, above which the limit-cycle oscillation and the associated DTC phase suddenly disappear?

Reply: Indeed, the limit cycle oscillations disappear at about 17.5 K, as was studied in our previous publication. In more detail, see Nat. Phys. 20, 631 (2024) and corresponding Supplementary Material. We connect the disappearance of the oscillations with the increased nuclear spin relaxation and reduced efficiency in the dynamical nuclear spin polarization by the electrons.

R#1 3). The sentence “The deviation causes a wide variety of phenomena ranging from synchronization to chaotic motion that depend on the frequency and depth of the modulation, being akin to the transformation from CTC to DTC” in the introduction is not very clear. Why is the transition from synchronization to chaotic motion akin to the transformation from CTC to DTC? Both CTC and DTC are ordered phases, but the chaotic motion is not. There are also some missing literatures on time crystals, e.g., the Rydberg time crystal recently observed in [arXiv:2305.20070, arXiv:2402.13657, etc.]. In particular, I noticed that [arXiv:2402.13657] studies phenomena very similar to the current paper, though in a different setup. It might be helpful to briefly comment on the difference between the works.

Reply: We thank the Reviewer for pointing us to these omissions. First, to avoid confusion, we have removed the part after the comma: “, being akin to the transformation from CTC to DTC.” Regarding the citations. We have now extended our citation list and commented on several papers in different parts of the paper.

We have first included citations of systems where self-sustained oscillations were observed previously (we are concentrating here on several most recent and relevant publications). These include Rydberg systems [X. Wu, et al., *Dissipative time crystal in a strongly interacting Rydberg gas*, *Nature Physics* 20, 1389 (2024).], as well as double quantum dots system [K. Ono and S. Tarucha, Nuclear-spin-induced oscillatory current in spin-blockaded quantum dots, *Phys. Rev. Lett.* 92, 256803 (2004)] and quantum Hall systems [G. Yusa, et al., Self-sustaining resistance oscillations: Electron-nuclear spin coupling in mesoscopic quantum hall devices, *Phys. Rev. B* 69, 161302 (2004); S. Hennel, et al., Nonlocal polarization feedback in a fractional quantum hall ferromagnet, *Phys. Rev. Lett.* 116, 136804 (2016).]

Furthermore, we have added a citation predicting the appearance of the Farey tree sequence and devil's staircase in an exciton-polariton structure [I. A. Ramos-Pérez, et al., Theory of optomechanical locking in driven-dissipative coupled polariton condensates, *Phys. Rev. B* 109, 165305 (2024).].

We have added citations to interacting Rydberg gas systems demonstrating fractional subharmonics and bifurcations, see [Yuechun Jiao, et al., Observation of time crystal comb in a driven-dissipative system, arXiv:2402.13112v1 (2024); Bang Liu, et al., Bifurcation of time crystals in driven and dissipative Rydberg atomic gas, arXiv:2402.13644v3 (2024); Bang Liu, et al., Higher-order and fractional discrete time crystals in floquet-driven rydberg atoms, arXiv:2402.13657v4 (2024)].

Finally, we cite the very recent similar work on CTC to DTC transition for a BEC [P. Kongkhambut, et al., Observation of a phase transition from a continuous to a discrete time crystal, *Reports on Progress in Physics* 87, 080502 (2024)] and a realization of a quasi-time-crystal state on spins in diamond [G. He, et al., Experimental realization of discrete time quasi-crystals (2024), arXiv:2403.17842].

R#1 4). In the experiment, the circularly polarized pump laser is used to drive the CTC and the DTC, while the linearly polarized probe field is used to probe the spin dynamics. I am curious why the power of the probe field (1 mW) is much larger than the pump field (0.3 mW)? Normally, the probe field needs to be sufficiently weak in the pump-probe technique, otherwise the probe field has a non-negligible influence on the dynamics. It would be helpful to comment on such a special choice of parameter regimes.

Reply: Indeed, this choice might seem unusual. However, our previous paper thoroughly studied power dependencies; see *Nat. Phys.* 20, 631 (2024) and corresponding Supplementary Material. In simple terms, the substantial difference in the wavelength between pump and probe and the effects of power on the FFT spectra dictate the choice. So, the pump power strongly affects the FFT peak positions and amplitude of the oscillations. The pump laser is fully absorbed by the system. The probe, in contrast, is applied in the transparency region; it does not affect the FFT peak positions and is chosen to be so strong to provide a high signal amplitude. This is convenient as we don't use modulation of the probe to amplify the signal by a lock-in amplifier, because a probe modulation applied to the system may lead to additional effects.

R#1 5). The authors should re-check the values of f_0/f_m in Fig. 2c. The ground frequency of the CTC is $f_0 = 0.1607$ Hz, such that the modulation frequency f_m used in Fig. 2b should give $f_0/f_m = 1, 0.93, 0.84, 0.81$. I guess the authors might use a different f_0 here, e.g., the one under the average laser polarization as they state in the caption. If so, I think it is quite confusing, because the gray and the orange curves in Fig. 2b collapse to each other, but $f_0/f_m \neq 1$ in Fig. 2c. They should either show the values I listed above or introduce a different notation for this distinct f_0 (e.g., f_0') and indicate its value. If there are similar issues elsewhere (e.g., the x-axis of Fig. 5), the authors should change them as well.

Reply: Yes, this is an issue that we didn't highlight enough. In the case of Fig. 2b,c the value of f_0 without the modulation (the natural frequency) is 0.1607 Hz. In that case, the circular light is applied permanently. With sinusoidal modulation of full depth ($\varepsilon = 1$), the degree of polarization varies between circularly and linearly polarized light so that, on average, the system sees elliptical light. This affects the position of the natural frequency, as is shown in Supplementary Figure 6, where the dependence on a constant degree of light polarization is measured. Additionally, a modulation frequency close to the natural frequency affects the position of the observed harmonics. This leads to an effect, where although we apply modulation with the $f_m = f_0$ of the unmodulated system (= 0.1607 Hz), the ratio f_m/f_0 of the modulated system is not equal to unity. We follow the reviewer's suggestion to mark the frequencies differently. We decided to mark the f_0 in the unmodulated case as \bar{f}_0 , while keeping the notation in the modulated case f_0 . We have changed the labeling in the corresponding places and added the following text: *"It is important to note that \bar{f}_0 is the system's natural frequency without modulation. If modulation is applied, the pump's average helicity is changed, renormalizing the natural frequency to f_0 ."*

Additionally, we have added the following sentence in the methods section: *"To detect the position of f_0 in the modulation case, we set the modulation frequency f_m to about $3.5\bar{f}_0$. In this way, the position of f_0 is unaffected by the proximity of the modulating frequency."*

R#1 6). The authors should provide more details on how they determine the edges of the entrainment ranges. The current argument based on the appearance of multiple Fourier components is somewhat qualitative. For example, is there a good order parameter that can be used to obtain the Arnold tongue in Fig. 3, and what is the definition of the error bars on the edges? I do not find a clear description of these details.

Reply: The determined position of the edges is purely qualitative in our case. It serves only demonstrative purposes. There are ways to make it quantitative, such as the correlation dimension D_2 . Supplementary Fig. 3 demonstrates that the correlation dimension changes its value from 2.5 to 1 by transitioning to the synchronized periodic state. Any deviation from $D_2 = 1$ would mark the end of the synchronization interval. Additionally, we provide an example of the automatized determination of the K -parameter, which can be used to describe the degree of periodicity of the time series [D. Toker, et al., "A simple method for detecting chaos in nature", *Communications Biology* **3**, 11 (2020)]. So, if K comes closer to 0, the system is more periodic, and if it comes closer to 1, it is more chaotic. Figure 1 in this reply demonstrates a calculation of the K -value for a simulated time series by stepping over the synchronization plateau. We were not taking the required fine steps in the experimental case, as it would have been unnecessarily time-consuming. Instead, we searched for the edges where the clean single FFT peak transitions to the peak with multiple neighboring harmonics. This gives a good indication of the frontier of the plateau.

Fig. 1. Top panel, simulated FFT in a small range around the synchronization plateau of $f_0/f_m = \frac{1}{2}$ demonstrating synchronization to the polarization modulation with $\varepsilon = 1$. In the bottom panel, K-parameter values for the corresponding time traces were calculated. K close to 1 suggests chaotic behavior, while K close to 0 indicates periodic behavior.

In our case, the error bars were determined through the frequency step used in the measurements. Their length is half the measurement step at the observed transition between fully synchronized and unsynchronized FFTs. For example, in the case of Fig. 3a, the edge of the plateaus for the full modulation depth at one side of the plateau would be in the middle of $f_0/f_m = 0.477$ and 0.484 , with the error bars being half of that interval. Indeed, this error bar can be reduced by making smaller steps. Still, it is sufficient to make the point of reduction of the synchronization interval by reduction of the modulation depth, as demonstrated in Fig. 3b.

We have clarified this situation in the main paper's figure 3 caption by adding the text: *"The border is set at the half-step size between the synchronized case with clean FFT peaks and the unsynchronized case with multiple side peaks in units of f_0/f_m . The error bars are correspondingly given by half-step sizes in the same units."*

R#1 7). To achieve a better agreement with the experiment, the modulation-depth parameter ε_{sim} needs to be rescaled, which is attributed to the off-resonant excitation in the current manuscript. Have the authors perform the observation in the near-resonant regime to justify this treatment and what is the qualitative dependence between ε_{sim} and the detuning?

Reply: In our case, the relation between the theoretical and experimental modulation depth is only qualitative. The ε_{sim} dependence on the external parameters is not trivial and depends on pump power density, pump laser wavelength, and modulation type. We have observed a much stronger effect by modulating the pump power instead of pump polarization. This allowed us to achieve areas where the neighboring Arnold tongues start to overlap, similar to the simulated case with $\varepsilon_{sim} = 0.75$ in Supplementary Fig. S4b. This case deserves further investigation in follow-up publications.

We have done several test measurements for the resonant case of the pump wavelength and observed self-sustained oscillations. However, the measurements become more demanding as one must take care of the transmitted/scattered pump laser light. In our case, we decided to keep the pump in non-resonant conditions to avoid additional modulations, which would be required for the lock-in technique. Overall, further experiments are needed and planned to specify the dependence of the modulation depth on the length of the synchronization plateaus.

R#1 8). In Fig. 5, the devil's staircase associated with $f_0/f_m = 2/3$ can be clearly identified in the calculation (Fig. 5b) but not in the experiment (Fig. 5a). Is there any specific reason for such a difference? I suppose that the entrainment range is still considerable for this stage. Besides, it would be helpful to indicate the chaotic region, e.g., do all regions between two neighboring entrainment ranges exhibit chaotic behaviors?

Reply: Following the suggestion of the Reviewers (see also point 2 of Reviewer 3), we decided to invest considerable time in remeasuring Fig. 5a with higher resolution (we additionally included the Fig. 5a in a stretched format, see the supplementary Fig. S7). Now, one can recognize the structure of the Farey tree sequence with more details. Also, the length of the plateaus does indeed decrease with increasing denominator. So, for the plateaus $1/2$, $3/5$, $2/3$, $3/4$, and $4/5$, one gets lengths of 0.059, 0.012, 0.018, 0.013, and 0.010 ± 0.001 in units of f_0/f_m , respectively. We observe deviations between experiment and theory in terms of absolute values of the plateau lengths. Still, considering the model's simplicity, the general structure is reproduced very well.

As for the chaotic regions, yes, all regions connected to the synchronization plateaus should demonstrate chaos. However, considering that the main staircase with the smallest denominator in the fractions gives the strongest amplitude, we experimentally observe the chaos only at the transition to the synchronization region $1/1$. One has to keep in mind that for any other region with multiple bifurcations, which are seen in the simulation close to each synchronization plateau, the signal's amplitude is drastically reduced. This makes it challenging to measure further regions experimentally as it would require much longer accumulation times.

We have discussed this point by adding: "Our simulation shows that such ranges with a very high number of bifurcations appear close to each transition to synchronization, see Figs. 5b and 5c. Due to the redistribution of the frequency components between multiple subharmonics, these are much harder to observe experimentally."

R#1 9). In general, the emergence of the DTC phase does not necessarily require the CTC phase. Have the authors observe (experimentally or theoretically) the DTC order in the regime where CTC does not exist?

Reply: We are grateful to the Reviewer for this comment. Indeed, we have done tests for experimental conditions where no self-sustained oscillations were observed (angle of the magnetic field to the sample 30°), and no CTC state is realized. However, once the polarization modulation was applied, we could detect oscillations with multiple subharmonics, representing the DTC state (see Fig. 2 in this report). This observation demands follow-up investigations on this specific phenomenon, which we will provide in follow-up publications.

Fig. 2. Left panel, time traces for the unmodulated pump (blue) and modulated pump (red). Polarization modulation is done with full depth, $\varepsilon = 1$. Right panel, the corresponding FFT spectra. The blue trace shows no peaks, while the red one demonstrates the strong peak at the driving frequency with multiple subharmonics at fractional frequencies.

Reviewer #2 (Remarks to the Author):

In the manuscript entitled "Exploring nonlinear dynamics in periodically driven time crystal: from synchronized to chaotic motion," the authors study deviations of the coupled electron-nuclear spin system in an InGaAs semiconductor from continuous driving by periodic modulation of the excitation polarization. The authors reveal several phenomena that depend on modulation frequency and depth. In particular, the width of these ranges depends on the polarization modulation depth, resulting in Arnold Tongues.

This work is interesting, but in my opinion, it is a natural continuation of previous works by the authors, so it seems to me that it is not within the scope of the journal in the sense that it aims to represent important advances of significance for specialists within each field. I then ask the authors to justify better why they believe the work merits publication in a high-impact journal. Furthermore, I call on you to consider the following points.

Reply: We thank the Reviewer for carefully reading the manuscript and assessing this work as interesting. While we agree that this work builds on our previous studies, we respectfully argue that it introduces critical advances beyond the earlier results. We believe these advances make the work highly relevant to the journal's readership for the following reasons:

- In the previous publication, we have considered an autonomous system, while here, the system becomes **non-autonomous**.
- It presents **new physics** by bridging the gap between classical nonlinear dynamics (e.g., synchronization, Arnold's tongues) and quantum-driven systems (electron spins interacting with nuclear spin bath).
- It offers a **robust solid-state platform** for studying time crystals, synchronization, and chaos, which have primarily been explored in atomic or optical systems.
- The results have potential implications for **quantum technologies**, including frequency standards and precision measurements.

R#2 2).- In the introduction, the authors spend large passages of the text discussing the phenomenon of oscillator synchronization; however, when the system is forced periodically, other mechanisms may be relevant, such as resonance and parametric resonance. In particular, the latter presents Arnold's tongues. I recommend that this point be clarified.

Reply: We appreciate the opportunity to clarify this point. It is correct that an Arnold's tongue and the effect of entrainment can appear in different systems as a manifestation of resonance or parametric resonance. Our system exhibits self-sustained oscillations due to intrinsic feedback mechanisms between the optically pumped electron spins and dynamically polarized nuclear spins. This intrinsic nonlinearity makes the system fundamentally different from a non-autonomous oscillator subject to external periodic driving. To address this point, we have revised the introduction to explicitly distinguish the mechanisms leading to resonance in non-autonomous systems (resonance and parametric resonance) from those in nonlinear systems with self-sustained oscillations.

The added sentences are reading now: *"It is essential to highlight the difference between the synchronization observed in nonlinear autonomous systems and another well-known phenomena, resonance and parametric resonance. The latter can be observed in periodically driven systems that do not demonstrate self-sustained oscillations. There is no synchronization, as the system does not have its own rhythm, and if the periodic driving is interrupted, the oscillation stops after some transition time [Pikovsky_book]."*

R#2 3) *.- Subhormonal instabilities correspond to the emergence of half-frequencies [initially studied by M. Faraday, Philos. Trans. R. Soc. London 52, 319 (1831), and see Landau L. D. and Lifshitz E. M., Mechanics, (Course of Theoretical Physics, Pergamon Press, 1976)], this should be better detailed in the text.

Reply: We thank the Reviewer for this remark and have incorporated a corresponding short discussion in the introduction. It reads: *“The effect of parametric resonance or subharmonic oscillations, initially observed by Faraday in periodically driven granular material [Faraday1831,Goldstein2018], represents a fundamental mechanism where a system oscillates at a fraction of the driving frequency. Landau and Lifshitz later formalized this phenomenon in the context of parametric resonance [Landau1976Mechanics], highlighting its universal occurrence. Unlike the classical Faraday instability, our results arise in a nonlinear auto-oscillating system, demonstrating a synchronization of higher orders and offering new insights into subharmonic dynamics.”*

R#2 4) *.- On the other hand, Figures 2b, 2c, and 4a show that complexity emerges from the quasi-periodic path. However, the analysis presented is not conclusive.

Reply: We appreciate the Reviewer’s concern about the lack of analysis and a keen observation regarding the emergence of the quasi-periodicity. Quasi-periodic states predominantly appear in the regions between large synchronization plateaus, where the system exhibits incommensurate frequencies. However, as indicated by the Devil’s Staircase structure, these regions are interspersed with smaller synchronization plateaus corresponding to higher-order rational frequency ratios.

To improve the analysis, we have now added a discussion on the fractal dimension of the observed structure, which was possible due to increased resolution in our new set of data, see Fig. 5a. In our case, the system exhibits a fractal structure with scaling behavior near criticality, similar to the well-known universality class described by Bak et al. “Mode-Locking and the Transition to Chaos in Dissipative Systems” Physica Scripta, Volume 1985, Number T9 (1985) (DOI 10.1088/0031-8949/1985/T9/007). Specifically, the observed fractal dimension of the Farey tree structure determined from the experimental data ($D' = 0.852 \pm 0.021$) and from the simulated data ($D' = 0.853 \pm 0.002$) aligns with the universal properties of mode-locking transitions and the Devil’s Staircase structure seen in dissipative systems. This suggests that the ENSS likely belongs to a broader universality class, sharing convergence and scaling properties akin to those found in circle maps and forced oscillators.

See also the discussion in the reply to the Reviewer 3, point 5, where we describe the additional text added to the paper.

R#2 5) *.- It needs to be clarified whether the phenomenon studied is local or is a spatially extended phenomenon where spatial behaviors can be observed. For example, chaotic behaviors are known through quasi-periodic routing [M.G. Clerc, and N. Verschuere, Phys. Rev. E. 88, 052916 (2013)]. Comment on this point.

Reply: We thank the Reviewer for raising this important point. In our system, spatial coupling is indeed present and is observed as a synchronization between remote donor centers, making the phenomena partially nonlocal. However, the primary focus of this study is on the temporal dynamics observed in an ensemble of donors, the laser spot includes millions of donors. Future investigations could explore the potential for spatially chaotic behavior in our system, inspired by studies such as Clerc & Verschuere [Phys. Rev. E. 88, 052916 (2013)].

We now have added this point to the outlook at the end of the paper, stating: *“The phenomena studied in this work are primarily driven by local dynamics arising from nonlinear feedback between*

electron and nuclear spins within an ensemble of donors. However, the spatial coupling between donors would introduce a degree of nonlocality, which can lead to spatial synchronization across neighboring donors. While our current analysis focuses on temporal dynamics, future work will explore potential spatial patterns in terms of partial synchronization [kuramoto2002, AbramsPRL04, Panaggio2015] or instabilities in the chaotic regime, similar to those described in spatially extended systems [Clerc & Verschueren, Phys. Rev. E. 88, 052916 (2013); Russomanno, Phys. Rev. B 108, 094305 (2023)].”

R#2 6) Minor comment

*.- The references [3] and [29] are the same.

Reply: We assume the Reviewer was referring to the citations [2] and [29]. However, in Ref. [2], we refer to the whole book of Pikovsky et al., while in Ref. [29], we specifically cite Chapter 7.

Reviewer #3 (Remarks to the Author):

The article by A. Greilich et al presents a study of a classical, non-linear system established by an electron spin interacting with nuclear spins - all in a semiconductor structure accessed optically. The authors build up on previous work, which revealed the presence of long-lived auto-oscillations in electronic polarization (considered a continuous time crystal). The new results presented in this article stem from periodically modulating the polarization of a laser that orients the electron spin along a fixed axis. A number of fascinating phenomena arise due to feedback between the electron and nuclear spins; for certain parameters the system exhibits synchronization ('entrainment'), and it transitions into chaos as the parameters are varied. The growth of the entrainment range with an increase in modulation depth is a particularly exciting finding. The data are backed by an appropriate theoretical analysis, and the article is very well-written and cohesive.

I definitely think that the article fits well into Nature Communications and that it will attract a lot of attention from physicists from various fields. I am supportive of a publication, but I think some revisions are in place. This is because certain claims made in the article can lead to a backlash, as well as because the microscopic model is not completely clear. I would be thankful if the authors could address the following points:

R#3 1) Equation 2 describes the evolution of the Overhauser field in terms of the electronic spin. It introduces a tensor of unspecified form, and a claim is made that $\hat{a}\mathbf{S}$ is a linear function of \mathbf{S} . Should this be understood as $\hat{a}\mathbf{S}=a\mathbf{S}$ where a is a scalar? Or is \hat{a} still a tensor, but a one which has components independent of \mathbf{S} ? I believe details are important here. \hat{a} certainly should not be a scalar: without strain, in a frame rotating with the electronic precession frequency, the hyperfine tensor averages out to a matrix with a single non-zero component, reducing the hyperfine interaction between the electron and nuclei from 'Heisenberg' to 'Ising' type. As authors know, the strain is the key ingredient here, as it introduces the effective non-collinear components to the hyperfine tensor, which couple z-components of the electronic polarization with x,y-components of the nuclear polarization. These interactions allow for the polarization transfer between the electron and the nuclei, and are therefore important to elaborate on. Moreover, as written in the manuscript 'The Overhauser field of the polarized nuclear spins, in general, is oriented not parallel to the average electron spin' which would not be consistent with Equation 2 if \hat{a} was a scalar quantity. It would be great if authors presented the complete form of \hat{a} at least in methods. This also goes for all other parameters used in the model. It's impossible to reproduce the numerics without these.

Reply: We appreciate the Reviewer's high evaluation of our work. We have kept the justification of the equation form quite short in this paper, as it was fully presented in Ref. [4] and corresponding supplementary material. We have now extended the Method section, where we explicitly write the form of the tensor.

R#3 2) Certain features of Fig. 5b are not reproduced in Fig. 5a. As an experimentalist, I appreciate how demanding task it was to collect the data behind Fig. 5a, and that improving the resolution along y-axis would force one to collect the data for extremely long times. However, that doesn't apply for a window of $f_0/f_m \in [0.65, 0.7]$: and there is a clear deviation of the data and simulation in that window. I believe without it, it is a little bit of a stretch to claim the observation of 'devil's staircase' structure (lines 65-68); surely it is supported by simulation, but not exactly by the data. Moreover, Fig. 5 misses a colorbar.

Reply: We thank the Reviewer for this question. It has motivated us to provide additional measurements, where we have taken much finer steps in the modulation frequency. We have replaced Fig. 5a and additionally included the Fig. 5a in a stretched format, see the supplementary Fig. S7. As we have argued for the Reviewer 1, one can now recognize the structure of the Farey tree sequence in much more detail. The length of the plateaus decreases with increasing denominator. So, for the plateaus $1/2$, $3/5$, $2/3$, $3/4$, and $4/5$, one gets lengths of 0.059, 0.012, 0.018, 0.013, and 0.010 ± 0.001 in units of f_0/f_m , respectively. There are some deviations from the theory in terms of absolute values of the plateau lengths. But, considering the model's simplicity, the general structure is reproduced very well. We have also included the color bar in Fig. 5.

R#3 3) In the final paragraph of the manuscript (lines 432-434), a statement is made that 'This state can be seen as a stable internal clock that can be implemented in a semiconductor chip'. The ENSS itself is not a good frequency standard; its model parameters will vary from sample to sample, and the frequency stability of the auto-oscillations is not really explored here (nor there is any reason to believe it would be comparable to technologically relevant devices). It is therefore misleading to suggest that such semiconductor chip could realize a 'stable internal clock'. The interesting bit is that ENSS can adopt a frequency of an external frequency generator – which I fully agree with. However, I find the idea to bring it up as a metrologically-relevant outlook a little bit odd. Firstly, the modulation frequencies to which the system locks are extremely low; resolving any frequency difference in a metrological scenario would take a prohibitively long time. Could authors be a little bit more specific which metrological applications do they have in mind? It is also okay to take down the relevance of this system to metrology - it will not diminish the importance of this article.

Reply: Upon reflection, we agree that this suggestion may stretch beyond the immediate implications of our work, especially given the frequency stability and sample variability issues you highlighted. We have therefore decided to remove the statement about the ENSS serving as a "stable internal clock" in a semiconductor chip and its relevance to metrology.

Instead, we have revised the conclusion to focus on the fundamental insights gained from the system's ability to adopt external frequencies and highlight the potential implications for understanding nonlinear dynamics and time-crystalline behavior. We believe this change will make the manuscript more accurate and aligned with the work's scope.

R#3 4) The system under study is fascinating, and it looks like there could be a number of in-depth experimental studies of Chaos and non-linear dynamics performed with it in the near future. I think the article would gain from some outlooks which hint towards what's going to be explored next, some

comments that draw a big picture. Beyond being a playground for chaos, it would be great to see this platform become technologically relevant. The authors suggested metrology, which I do not think is a viable route forward, but maybe there are other potential applications?

Reply: We thank the Reviewer for the enthusiasm and insightful suggestions. We are equally excited about the future potential of our system for studying chaos, nonlinear dynamics, and related phenomena. We have taken your suggestion to extend the conclusion section and included a broader outlook highlighting possible directions for future research.

In particular, we have added:

1. Exploration of Spatial Coherence and Nonlocality:
 - We emphasize the opportunity to investigate spatial synchronization and coherence effects mediated by electron spin diffusion, which could further reveal the interplay between local and nonlocal dynamics in our system.
2. Potential Applications in Quantum Phenomena:
 - We highlight the potential for this system to explore quantum effects in driven-dissipative environments, such as quantum synchronization and coherence, which could inspire new applications in quantum technology, beyond metrology.

These additions aim to provide a broader perspective on how the system could evolve into a versatile platform for fundamental studies and potential technological applications.

We hope these revisions address the reviewer's suggestion and provide the "big picture" outlook asked for.

R#3 5) As a huge fan of renormalization, I have to ask this out of pure curiosity (by no means do I urge authors to include this discussion in the article): cascades of bifurcations usually exhibit convergence properties described by Feigenbaum constants (at least in unimodal maps) - what's expected in that respect from the ENSS under study?

Reply: We thank the reviewer for this fascinating question about the applicability of Feigenbaum constants to the ENSS system. Indeed, cascades of bifurcations, such as period-doubling, are often characterized by Feigenbaum constants in unimodal maps. In our case, the system exhibits a fractal structure with scaling behavior near criticality, similar to the well-known universality class described by Bak et al. "Mode-Locking and the Transition to Chaos in Dissipative Systems" *Physica Scripta*, Volume 1985, Number T9 (1985) (DOI 10.1088/0031-8949/1985/T9/007). Specifically, the observed fractal dimension of the Farey tree structure determined from the experimental data ($D' = 0.852 \pm 0.021$) and from the simulated data ($D' = 0.853 \pm 0.002$) aligns with the universal properties of mode-locking transitions and the Devil's Staircase structure seen in dissipative systems. This suggests that the ENSS likely belongs to a broader universality class, sharing convergence and scaling properties akin to those found in circle maps and forced oscillators.

While our current work focuses on the general aspects and phenomena observed in electron-nuclear spin systems under periodic driving, the emergence of complexity—particularly the transition from quasi-periodicity to chaos—suggests a deeper connection to universality classes observed in nonlinear systems. Given the simplicity and clarity of the underlying equations, we believe this system represents an excellent opportunity for theoreticians specializing in **renormalization group analysis** to explore these dynamics further. A full theoretical study could reveal fixed points, scaling relationships, and universality properties governing the transitions we observe. Such an analysis would not only deepen the understanding of our system but also situate it within a broader theoretical framework of mode-locking, torus breakdown, and chaos transitions. We hope that our results can inspire future studies in this direction.

We have additionally extended the discussion part on the determined values of the fractal dimension and its connection with the universal class of driven dissipative systems by adding: "*In Ref.*

[Jensen_PRA1984], the authors studied the scaling law underlying the Arnold tongues. Similar to their results, the observed fractal structure, in our case, is related to the structure of the self-similar Cantor set. Using the experimentally observed Farey tree sequence with major plateaus and gaps between them, we arrive at the fractal dimension value of $D^1 = 0.852 \pm 0.021$, while the simulated data, with the possibility to zoom into the staircase structure, delivers $D^1 = 0.853 \pm 0.002$, see Methods. These values are close to 0.87 expected for the complete devil's staircase [Jensen_PRL1983] and align with the universal properties of mode-locking transitions and the devil's staircase structure seen in dissipative systems [Per_Bak_1985] as well as in Refs. [Brown84, Baums89, Wu2022]. This suggests that the ENSS likely belongs to a broader universality class, sharing convergence and scaling properties akin to those found in circle maps and forced oscillators.]

Response to reviewers of NCOMMS-24-45342A, "Exploring nonlinear dynamics in periodically driven time crystal: from synchronized to chaotic motion", by A. Greilich *et al.*
First and foremost, we would like to thank the Reviewers for their positive final evaluation. Please find below a point-by-point response to all comments.

Reviewer #1 (Remarks to the Author):

I thank the authors for the detailed responses to my comments and for careful revisions of the manuscript. With this, I can recommend publication of the paper in Nature Communications.

P.S., maybe one more comment on Q5: The notation \bar{f}_0 is suggested for the frequency under the averaged pumping. Since the authors want to keep the notation f_0 for such an averaged value, they can also choose another more suitable notation for the natural frequency, e.g., f_{0n} ("n" for "natural") or f_{0i} ("i" for "intrinsic").

Reply: We thank the Reviewer for their positive evaluation of the paper and recommendation for the publication. We have also decided to follow the comment and changed the notation from \bar{f}_0 to f_0^{nat} in the Fig.1 as well as in the corresponding places in the text.

Reviewer #2 (Remarks to the Author):

In the revised version, the authors have incorporated all comments from the previous peer review reports and adequately addressed most comments and suggestions. In particular, the authors more adequately justify using a semi-classical approach for their study. The disappearance of the limit cycle is clarified in the supplementary material. The state of the art has been significantly improved. Some technical aspects of the experiment, such as the power used for the light beam, have been clarified. The difference between synchronization and sustained oscillations has been better clarified. The parametric instabilities have also been clarified.

My biggest difficulty is the fulfillment of the Nature Communications criteria. The authors present a work supported by previous work, where the main novelty is considering a new platform based on a solid state of matter and its possible applications. The possibility of new applications undoubtedly gives interest to the present work on a phenomenon well-known to the nonlinear science community. Overall, the quality of the paper has improved significantly, and it is therefore recommended for publication.

Reply: We appreciate the positive evaluation of the paper and recommendation for the publication.

Reviewer #3 (Remarks to the Author):

The authors have responded satisfyingly to all the points I raised, and the quality of the manuscript has improved. I recommend its publication in its current form.

Reply: We thank the Reviewer for their positive final evaluation of the paper and recommendation for the publication.

The manuscript “Exploring nonlinear dynamics in periodically driven time crystal: from synchronized to chaotic motion” studies optical response of a periodically driven, nonlinear solid-state system. The study is based on their previous work [Nat. Phys. 20, 631 (2024)], which identifies a persistent auto-oscillation of the polarization of the probe field passing through an InGaAs semiconductor subjected to a continuous external driving. The generalization to the periodic driving regime gives rise to a number of new phenomena as key novelties of the current work, e.g., the frequency entrainment to fractional values of the driving frequency, which can be regarded as a discrete time crystal (DTC).

The authors carry out a systematic investigation of these new phenomena. They measure the time trace of the signal by varying the frequency and the depth of the periodic modulation, and analyze the results in terms of the Fourier spectrum, Arnold tongue, and bifurcation diagram. These results are of interest to the physics community, as well as to broader readers in complex systems, nonlinear sciences, and electronics. Therefore, I tend to think that the work is suitable for publication in *Nat. Commun.* However, before I can make the final decision, there are several critical points to be clarified and further elaborated on.

The detailed comments and questions are listed below:

1. The observed nonlinear dynamics is essentially classical, so I am wondering if the system can be extended to the quantum regime and how, e.g., what is the role of quantum correlation and entanglement and how to measure them? The authors develop a phenomenological model to describe the nonlinear spin dynamics, which has a nice agreement with the experiment. Is it also possible to develop a first-principle, fully quantized many-body model, even it may not be easy to solve? In short, I think it would be better to describe how to go beyond the classical regime, as conventional bifurcations and chaos are well understood in nonlinear sciences.
2. Related to comment 1, I noticed that the sample is already put in a low-temperature environment, where the quantum mechanical effect should play a role. Does the system have a critical temperature, above which the limit-cycle oscillation and the associated DTC phase suddenly disappear?
3. The sentence “The deviation causes a wide variety of phenomena ranging from synchronization to chaotic motion that depend on the frequency and depth of the modulation, being akin to the transformation from CTC to DTC” in the introduction is not very clear. Why is the transition from synchronization to chaotic motion akin to the transformation from CTC to DTC? Both CTC and DTC are ordered phases, but the chaotic motion is not. There are also some missing literatures on time crystals, e.g., the Rydberg time crystal recently observed in [arXiv:2305.20070, arXiv:2402.13657, etc.]. In particular, I noticed that [arXiv:2402.13657] studies phenomena very similar to the current paper, though in a different setup. It might be helpful to briefly comment on the difference between the works.

4. In the experiment, the circularly polarized pump laser is used to drive the CTC and the DTC, while the linearly polarized probe field is used to probe the spin dynamics. I am curious why the power of the probe field (1 mW) is much larger than the pump field (0.3 mW)? Normally, the probe field needs to be sufficiently weak in the pump-probe technique, otherwise the probe field has a non-negligible influence on the dynamics. It would be helpful to comment on such a special choice of parameter regimes.
5. The authors should re-check the values of f_0/f_m in Fig. 2c. The ground frequency of the CTC is $f_0 = 0.1607$ Hz, such that the modulation frequency f_m used in Fig. 2b should give $f_0/f_m = 1, 0.93, 0.84, 0.81$. I guess the authors might use a different f_0 here, e.g., the one under the average laser polarization as they state in the caption. If so, I think it is quite confusing, because the gray and the orange curves in Fig. 2b collapse to each other, but $f_0/f_m \neq 1$ in Fig. 2c. They should either show the values I listed above or introduce a different notation for this distinct f_0 (e.g., \tilde{f}_0) and indicate its value. If there are similar issues elsewhere (e.g., the x-axis of Fig. 5), the authors should change them as well.
6. The authors should provide more details on how they determine the edges of the entrainment ranges. The current argument based on the appearance of multiple Fourier components is somewhat qualitative. For example, is there a good order parameter that can be used to obtain the Arnold tongue in Fig. 3, and what is the definition of the error bars on the edges? I do not find a clear description of these details.
7. To achieve a better agreement with the experiment, the modulation-depth parameter ϵ_{sim} needs to be rescaled, which is attributed to the off-resonant excitation in the current manuscript. Have the authors perform the observation in the near-resonant regime to justify this treatment and what is the qualitative dependence between ϵ_{sim} and the detuning?
8. In Fig. 5, the devil's staircase associated with $f_0/f_m = 2/3$ can be clearly identified in the calculation (Fig. 5b) but not in the experiment (Fig. 5a). Is there any specific reason for such a difference? I suppose that the entrainment range is still considerable for this stage. Besides, it would be helpful to indicate the chaotic region, e.g., do all regions between two neighboring entrainment ranges exhibit chaotic behaviors?
9. In general, the emergence of the DTC phase does not necessarily require the CTC phase. Have the authors observe (experimentally or theoretically) the DTC order in the regime where CTC does not exist?